

# Experimental calibration assessment of a MPLNET/Micro-Pulse Lidar system in comparison with EARLINET lidar measurements for aerosol optical properties retrieval

5    Carmen Córdoba-Jabonero[1]*, Albert Ansmann[2], Cristofer Jiménez[2], Holger Baars[2], María-Ángeles López-Cayuela[1], and Ronny Engelmann[2]

[1]Instituto Nacional de Técnica Aeroespacial (INTA), Atmospheric Research and Instrumentation Branch, Torrejón de Ardoz, 28850-Madrid, Spain
[2]Leibniz Institute for Tropospheric Research (TROPOS), Leipzig, Germany

10   *Correspondence to*: Carmen Córdoba-Jabonero (cordobajc@inta.es)

**Abstract.** Simultaneous observations of a polarized Micro-Pulse Lidar (P-MPL) system, currently operative within MPLNET (NASA Micro-Pulse Lidar Network), with two referenced EARLINET (European Aerosol Research Lidar Network) lidars, running at Leipzig site (Germany, 51.4ºN 12.4ºE, 125 m a.s.l.), were performed during a comprehensive two-month field campaign in summer 2019. A calibration assessment regarding the overlap (OVP) correction of the P-MPL signal profiles and its impact in the retrieval of the optical properties is achieved, describing also the experimental procedure used. The optimal lidar-specific OVP function for correcting the P-MPL measurements is experimentally determined, highlighting that the OVP function as delivered by the P-MPL manufacturer cannot be used. Among the OVP functions examined, the averaged one between those obtained from the comparison of the P-MPL observations with those of the other two referenced lidars seems to be the best proxy at both near- and far-field ranges. In addition, the impact of the OVP function in the accuracy of the retrieved profiles of the total particle backscatter coefficient (PBC) and the particle linear depolarization ratio (PLDR) is examined. First, the volume linear depolarization ratio (VLDR) profile is obtained and compared to the reference lidars, showing it needs to be corrected by a small offset value within a good accuracy. Once P-MPL measurements are optimally OVP-corrected, the PBC profiles (and hence the PLDR ones) can be derived using the Klett-Fernald approach. In addition, an alternative method based on the separation of the total PBC into their aerosol components is presented in order to estimate the total particle extinction coefficient (PEC) profile, and hence the Aerosol Optical Depth, from elastic P-MPL measurements. A dust event as observed at Leipzig in June 2019 is used for illustration. In overall, an adequate OVP function is needed to be determined in a regular basis to calibrate the P-MPL system in order to derive suitable aerosol products.

## 1   Introduction

Active remote sensing are an excellent tool for vertical monitoring of the atmosphere. In particular, aerosol lidar systems have demonstrated to be a suitable instrumentation for aerosol and cloud profiling in
both the troposphere and stratosphere (e.g., Amiridis et al., 2015; Baars et al., 2019). Tropospheric aerosols are usually confined up to 7-8 km height under aerosol intrusion conditions (e.g., Mattis et al.,



2008; Pappalardo et al., 2013); otherwise, they are mostly concentrated in the ABL (around less than 1.5 km height). Indeed, lidar systems are widely used due to their high vertical spatial and temporal resolution.

Ground-based lidar networks are widely operative within the GAW (Global Atmospheric Watch) Aerosol LIdar Observations Network (GALION); among them, there are those extended at continental scales, as EARLINET (European AeRosol LIdar NETwork, www.earlinet.org; Pappalardo et al., 2014), which belongs also to the Aerosol Cloud and Trace Gases Research Infrastructure (ACTRIS, www.actris.eu), AD-NET (Asian Dust and aerosol lidar observation network, www-lidar.nies.go.jp/AD-Net; Sugimoto et

al., 2008), and LALINET (a.k.a. ALINE, Latin American Lidar NETwork, www.lalinet.org; Barbosa et al., 2014). In addition, there are other aerosol networks like MPLNET (Micro-Pulse Lidar NETwork, mplnet.gsfc.nasa.gov; Welton et al., 2001), and PollyNET (POrtabLe Lidar sYstem NETwork, polly.tropos.de; Baars et al., 2016), whose sites are distributed around the world.

The use of polarization lidar options is increasing, since lidar depolarization measurements allow an

improved typing of aerosols (dust, marine aerosol, anthropogenic pollution, volcanic ash, biomass burning, pollen, …) as well as the separation of the optical properties (backscatter, extinction) of particle components within complex aerosol mixtures with vertical resolution (i.e., Ansmann et al., 2011; Burton et al., 2014; Yu et al., 2015; Sicard et al., 2016; Bohlmann et al., 2019). Therefore, new and promising methods based on the particle depolarization ratio were developed and used to derive aerosol profiles in

terms of particle mass concentration, separately for the coarse and fine modes (i.e., Mamouri and Ansmann, 2017), in addition to estimate both the cloud-condesation nucleii (CCN) and ice-nucleating particle (INP) concentrations (i.e., Mamouri and Ansmann, 2016).

The atmospheric lidar scanning provides an accurate characterization at all ranges; however, lidar systems present an incomplete response in the near-range observational field due to the partial intersection of the

field-of-view between the transmitter and the receiver for both the biaxial and coaxial lidar configurations. Therefore, lidar signal profiles must be corrected by this near-field loss of signal, that is, the overlap (OVP) correction (Wandinger and Ansmann, 2002); hence, the main concern is focused on their OVP calibration. The full-OVP height is different depending on the lidar system (e.g., Wandinger et al., 2016).

For the last two decades, the Micro-Pulse Lidar (MPL) (Campbell et al., 2002; Welton et al., 2002) has been the operative system within MPLNET; since a few years a polarized MPL version (P-MPL) is the standard lidar system. Both MPL and P-MPL observations have been widely performed for continuous monitoring of aerosols and clouds. In particular, MPL/P-MPL measurements were used for Atmospheric Boundary Layer (ABL) height retrievals (Lewis et al., 2013; Adame et al., 2015), for detection and

characterization of both cirrus clouds (Lewis et al., 2016; Córdoba-Jabonero et al., 2017) and Polar Stratospheric Clouds (PSC) (Campbell et al., 2008; Córdoba-Jabonero et al., 2013), for depolarization-based separation of the optical properties of different aerosol mixtures (Córdoba-Jabonero et al., 2018), for mass concentration estimation in comparison with forecast model simulations (Córdoba-Jabonero et al., 2019), for precipitation intensity determination (Lolli et al., 2018), and for assessment of the radiative

effect of aerosols and cirrus clouds (Campbell et al., 2016; Córdoba-Jabonero et al., 2020a, 2020b), among others. Those works have demonstrated a good MPL performance in aerosol/cloud research.



However, the P-MPL system needs to be well characterized in terms of the backscattered lidar signal detected by both depolarization channels of the instrument (Flynn et al., 2007) in order to retrieve plausible aerosol optical properties.

The P-MPL is an elastic coaxial single-wavelength (532 nm) system and, differing from older MPL versions (Campbell et al., 2002; Welton et al., 2002), incorporates depolarization capabilities (Flynn et al., 2007). It can operate in routine continuous (24/7) mode. From an instrumental point of view, the principal disadvantage is reaching the full-OVP height at relatively high altitudes (typically at 4-6 km height; Campbell et al., 2002), being particularly relevant for tropospheric aerosol research. Then, a

qualified calibration must be performed in these systems. In this framework, an experimental campaign was planned at the EARLINET Leipzig site (Germany, 51.4ºN 12.4ºE, 125 m a.s.l.), and devoted to compare P-MPL observations simultaneously with reference well-calibrated lidar measurements, and hence to determine the required P-MPL calibrations.

The aim of this work is fourfold: 1) to achieve an OVP calibration of the P-MPL system, i.e., to estimate

the experimental OVP function for correcting the P-MPL measurements; 2) to evaluate the volume linear depolarization ratio (VLDR), which is a lidar-derived parameter independent of OVP calibration; 3) to determine the P-MPL calibration-induced effects on the retrieval of optical properties, both the height-resolved particle backscatter coefficient (PBC) and particle linear depolarization ratio (PLDR); and 4) to present an alternative method based on the separation of the PBC into their aerosol components (for

instance, as applied to a dust event) in order to estimate the vertical profile of the particle extinction coefficient (PEC) (and also the columnar total extinction, i.e., the Aerosol Optical Depth, AOD) from elastic P-MPL measurements. **Section 2** introduces the methodology for that purpose, where the field campaign performed, the P-MPL and reference lidar systems used and the data analysis of the experimental approaches applied are particularly described: the experimental procedure for accurately

characterizing the OVP function of the P-MPL systems, the correction of the VLDR, and the determination of the optical properties. Results are presented in **Section 3**, regarding the experimental estimation of the OVP function (error processing is described in **Annex A**), the evaluation of the VLDR, and the retrieval of the particle optical properties. A dust case as observed during the field campaign is used for that purpose. Main conclusions are presented in **Section 4**.

## 2 Methodology

### 2.1 Field campaign

A field campaign was performed at the EARLINET station of Leipzig, Germany (51.35ºN 12.43ºE, 125 m a.s.l.), managed by the Leibniz Institute for Tropospheric Research (TROPOS), for 6 weeks in June-July 2019 in order to mainly calibrate a P-MPL system with a special emphasis on the OVP correction.

The lidar system used was the MPL44245 unit (Sigma Space Corp.) routinely operating at the MPLNET/El Arenosillo station at Huelva, Spain (ARN/Huelva, 37.1ºN 6.7ºW, 40 m a.s.l.), which is managed by the Spanish Institute for Aerospace Technology (INTA). Both stations are also AERONET (AErosol RObotic NETwork, aeronet.gsfc.nasa.gov) sites, accomplishing the requisite for co-location of both networks. For the campaign, the ARN/Huelva P-MPL was temporarily deployed at Leipzig to be



compared against two EARLINET lidars routinely operative in this station. Those are the Polly (POrtabLe Lidar sYstem; Althausen et al, 2009; Engelmann et al., 2016) and the MARTHA (Multiwavelength Tropospheric Raman lidar for Temperature, Humidity, and Aerosol profiling; Jiménez et al., 2018) systems, which were used as reference since these lidars are well characterized with respect to EARLINET quality assurance standards (e.g., Böckmann et al., 2004; Pappalardo et al., 2004;

Freudenthaler et al., 2008; Pappalardo et al.,2014; Wandinger et al., 2016; Belegante et al., 2016; Bravo-Aranda et al., 2016; Freudenthaler et al., 2016).

### 2.2 Lidar systems

#### 2.2.1 Polarized Micro-Pulse Lidar (P-MPL)

The P-MPL system (Sigma Space corp., v. MPL-4B) is the standard lidar currently operating within
MPLNET. It is an elastic lidar in coaxial configuration with depolarization capabilities operating in full-time (24/7) mode. Among the principal optical features, the laser emission at 532 nm, with a pulse energy of 6-8 μJ and a repetition rate of 2500 Hz, is registered back by a unique avalanche photodiode detector (APD), and the receiver system presents a field-of-view (FOV) of 80 μrad full angle and the telescope diameter is 18 cm wide (Sigma Space Corp., MPL system information handbook, 2018). P-MPL vertical
profiles are routinely acquired with 1-min integrating time and 15-m vertical resolution (in particular, for the ARN/Huelva P-MPL system) up to 30 km height. Main instrumental features of the P-MPL system are shown in **Table 1**.

**Table 1: Main instrumental features of the lidar systems.**

| Lidar system | P-MPL | Polly | MARTHA |
|---|---|---|---|
| Routine operation | 24/7 | 24/7 | Supervised |
| Lidar Networks | MPLNET | EARLINET | EARLINET |
| Transmitter properties | | | |
| Wavelength (nm) | 532 | 532 (*) | 532 (*) |
| Energy/pulse (mJ) | 0.006-0.008 | 400 | 1000 |
| Pulse frequency (Hz) | 2500 | 20 | 30 |
| Eye-safety | Yes (ANSI Class II) | No | No |
| Receiver properties | | | |
| Telescope diameter (cm) | 18 | 30 | 80 |
| Telescope focal length (m) | 2.23 | 0.89 | 9 |
| FOV (μrad full angle) | 80.4 | 1000 | 500 |
| Depolarization | Yes | Yes | Yes |
| Raman detection | No | Yes | Yes |

(*) Used in this study.

A schematic optical configuration of the MPL-4B version is shown in Flynn et al. (2007; see their Fig. 1). The laser light is alternatively transmitted linearly and circularly polarized to the atmosphere by switching between two retardation modes of a liquid crystal retarder (LCR). The corresponding backscattered light
to those two polarized states by passing through a beam splitter to the single APD is registered in



dependence of the polarizing or depolarizing atmospheric particles leading to the suppression or not, respectively, of the orthogonally-detected signal w.r.t. the transmitted one into the single APD. Those two polarized signals are semi-simultaneously detected by alternatively switching in the basis of 50%/50% the LRC polarization mode (LCR switching time of 133 μs) within every integrating minute. That is, those

two signals are alternatively detected by the same APD, being recorded in two polarized 'channels': the 532-nm cross-signal ($P_{cross}$) and the 532-nm co-signal ($P_{co}$). Therefore, since no potentially existing efficiency or alignment differences are between those two 'channels' (as used a single APD), no corrections for these effects are required, as it is typically needed for ordinary two-channel polarization lidars. The measured lidar signal in those two polarized-channels is used to derive the P-MPL total range-

corrected signal (RCS), $P^{MPL}$, following the methodology as described in Flynn et al. (2007), that is, $P^{MPL} = P_{co} + 2\,P_{cross}$. More details on P-MPL signal correction and data processing can be found in Córdoba-Jabonero et al. (2018). Among the required instrumental P-MPL calibrations (Campbell et al., 2002; Welton et al., 2002), the OVP correction is a concerning issue, since the typical full-OVP height is reached at rather high altitudes (usually at 4-5 km height), affecting thus the aerosol profiles at ranges in

the overall boundary layer and part of the troposphere. Therefore, after purchase, the P-MPL system is delivered with an original OVP function as provided by the manufacture company (Sigma Space Corp.), which, however, must be re-evaluated with time. Indeed, one of the goals of this work is to show the experimental procedure to obtain a new OVP calibration for the P-MPL lidar as compared to the original one, as will be exposed in **Sect. 2.3.1**, together to examine their effects in the retrieval of the optical

properties.

### 2.2.2    POrtabLe Lidar sYstem (Polly)

The EARLINET Polly (POrtabLe Lidar sYstem) lidars are sophisticated, automated Raman-polarization lidar systems for scientific purpose, but with the advantage of an easy-to-use and well-characterized instrument with same design, same automated operation, and same centralized data processing delivering

near-real-time data products. Polly systems have been developed and constructed at TROPOS with international partners since 2002 (Engelmann et al., 2016). All Polly lidar systems are designed for automatic and unattended operation in 24/7 mode. Meanwhile 12 Polly lidar systems are distributed around the globe (e.g., Baars et al., 2016). The Polly lidar system used as a reference in this comparison analysis, is the first one of the Polly family (Engelmann et al., 2016), which was substantially upgraded in

2016 (v. Polly_1v2). It emits linearly polarized light at 532 nm with 5 receiver channels: the elastically backscattered light at 532 nm, the cross-polarized light at 532 nm, the co-polarized light at 532 nm, the rotational-Raman scattered light near 532 nm, and the vibrational-rotational Raman scattered light at 607 nm. Its full-OVP is reached at around 300-500 m height, and thus preferred for the P-MPL OVP calibration purpose. Profiles of the Polly range-corrected signal, $P^{Polly}$, are routinely derived by using

sample settings with 7.5-m vertical resolution and 30-sec temporal integration. The main instrumental features of the Polly system are shown in **Table 1**.

### 2.2.3    Multiwavelength Atmospheric Raman lidar for Temperature, Humidity, and Aerosol profiling (MARTHA)

The second EARLINET lidar, which is used as a reference in this work, is the dual receiver field-of-view

(RFOV) Multiwavelength polarization/Raman lidar for Temperature, Humidity, and Aerosol profiling (MARTHA) (Mattis et al., 2008; Schmidt et al., 2013, Jimenez et al., 2019). MARTHA has a powerful laser, transmitting in total 1 J per pulse at a repetition rate of 30 Hz, with an 80-cm telescope diameter, being thus well designed for tropospheric and stratospheric aerosol observations. This lidar system measures Raman signals at 532 nm ($P^{MARTHA}$, which is that used in this work) and 607 nm and the

polarization-sensitive 532-nm backscatter signals at two RFOVs so that, besides aerosol profiles, cloud microphysical properties can be retrieved from measured cloud multiple scattering effects. MARTHA can provide the 532-nm particle depolarization ratio as measured with the smaller RFOV, and also the 355-, 532-, and 1064-nm particle backscatter coefficients and the 355- and 532-nm extinction coefficient profiles with their corresponding lidar ratio profiles. For this large telescope (and a selected receiver FOV

of 0.5 mrad) the overlap between laser beam and receiver FOV is complete around 2000 m height. The overlap profile of this laboratory lidar is very stable. The main instrumental features of the MARTHA system are shown in **Table 1**.

### 2.3  Experimental estimation of the overlap (OVP) function for P-MPL systems

The overlap (OVP) function, $F_{OVP}$, is used to correct the P-MPL RCS profiles, $P^{MPL}(z)$, at near-field

altitudes, that is,

$$P^{MPL}_{OVP}(z) = P^{MPL}(z) \Big/ F_{OVP}(z), \qquad (1)$$

where $P^{MPL}_{OVP}(z)$ represents the overlap-corrected P-MPL RCS profiles.

In this work, the experimental procedure to obtain $F_{OVP}$ is based on the comparison of the $P^{MPL}(z)$ to either the Polly RCS profiles, $P^{Polly}(z)$, or the MARTHA ones, $P^{MARTHA}(z)$, which are both used as

reference under mostly clean and clear conditions. The Polly and MARTHA lidars present the advantage in contrast to P-MPL system that the OVP function can be experimentally determined using their Raman channels (Wandinger and Ansmann, 2002). The P-MPL overlap function is thus calculated in terms of the ratio between the P-MPL and Polly/MARTHA RCS profiles, i.e.,

$$F_{OVP}(z) = P^{MPL}(z)/P^{ref}(z), \qquad (2)$$

where $P^{ref}(z)$ denotes the reference RCS profiles as obtained from either Polly, $P^{Polly}(z)$, or MARTHA, $P^{MARTHA}(z)$, measurements. Both sets of RCS profiles are normalized at a given height (higher than the OVP altitude range), $z_{norm}$, and then $F_{OVP}(z)$ can be derived using **Eq. 2**. In particular, the full-OVP is obtained at the normalization height $z_{norm} = 9.5$ km a.g.l., being $F_{OVP}(z) = 1$ at $z \geq z_{norm}$. Errors associated to the estimation of $F_{OVP}(z)$ using this experimental approach are described in **Annex A**. Lidar

observations performed under relatively clean conditions at the Leipzig station were used for the P-MPL OVP calibration.

### 2.4  Determination of the aerosol optical properties

#### 2.4.1    Retrieval of the particle backscatter coefficient, and both the volume and particle linear depolarization ratios


Once the OVP-corrected RCS is obtained from **Eq. 1**, the particle backscatter coefficient (PBC), $\beta_p$ (km$^{-1}$ sr$^{-1}$) can be derived applying the Klett-Fernald (KF) algorithm (Fernald, 1984; Klett, 1985) by constraining the lidar ratio (LR, extinction-to-backscatter ratio) with the AERONET Aerosol Optical Depth (AOD) (elastic KF solution); hence, an effective LR, $S_a^{eff}$, is also obtained after convergence. The particle linear depolarization ratio (PLDR), $\delta_p$, can be determined as follows,

$$\delta_p = \frac{R\,\delta^V\,(1+\delta_{mol}) - \delta_{mol}\,(1+\delta^V)}{R\,(1+\delta_{mol}) - (1+\delta^V)} \tag{3}$$

where $R$ is the backscattering ratio ($=\frac{(\beta_m+\beta_p)}{\beta_m}$, being $\beta_m$ the molecular backscattering coefficient), $\delta^V$ is the volume linear depolarization ratio (VLDR), and $\delta_{mol}$ is the molecular depolarization ratio ($\delta_{mol} = 0.0037$ for P-MPL systems; Behrendt and Nakamura, 2002). The PLDR is a lidar parameter widely used for defining the aerosol type (Burton et al., 2012; Gross et al., 2013), and for discriminating the particle

size mode in some aerosol mixtures (Mamouri and Ansmann, 2017; Córdoba-Jabonero et al., 2018), among others. The determination of PBC is mainly depending on the OVP correction, as will be discussed in **Sect. 3.3**, and hence, the PLDR is also affected by OVP as well. Therefore, a good knowledge of the OVP function for the specific P-MPL system is also needed to obtain high-quality PBC and PLDR profiles.

The volume linear depolarization ratio (VLDR), $\delta^V$, can be determined in relation with the P-MPL depolarization ratio, $\delta^{MPL}$ (Mishchenko and Hovenier, 1995; Gimmestad, 2008). Following Flynn et al. (2007), $\delta^V$ can be easily expressed as

$$\delta^V = \frac{\delta^{MPL}}{\delta^{MPL}+1} = \frac{P_{cross}}{P_{co}+P_{cross}}, \tag{4}$$

where $\delta^{MPL}$ is defined as the ratio between $P_{cross}$ and $P_{co}$, which are the two polarized RCS as described

in **Sect. 2.2.1**. Since the OVP correction is equally applied to both those signals, the VLDR is unaffected by the OVP calibration. Hence, the VLDR for the P-MPL system was examined in comparison with that derived from Polly lidar measurements. All those variables are height-resolved, but the altitude dependence is omitted for simplicity. A dust case occurring for the night on 29-30 June 2019 at the Leipzig station is selected for that purpose.

**2.4.2    Estimation of the particle extinction coefficient from elastic P-MPL measurements**

The particle extinction coefficient (PEC) is also a height-resolved variable. However, its estimation from elastic lidar observations is a relevant concern due to the indetermination in solving the lidar equation for elastic systems. Usually, a KF solution can be derived by assuming an effective LR, $S_a^{eff}$, which is a constant in height value (see **Sect. 2.4.1**).

An alternative simple method is introduced in order to estimate the PEC profiles, $\sigma_p(z)$, from elastic P-MPL measurements without assuming a constant LR. That approach is based in the combination of the POLIPHON algorithm (Mamouri and Ansmann, 2017) with elastic P-MPL measurements, as described in Córdoba-Jabonero et al. (2018), and was similarly used in Giannakaki et al. (2020), showing the potential of elastic and polarized lidars for vertical extinction retrieval. A dust case study observed at Leipzig in



June 2019 is used for illustration of that methodology. First, the KF-derived PBC profile, $\beta_p(z)$, together with the PLDR one, $\delta_p(z)$, are used to separate the total $\beta_p$ into the backscatter coefficients corresponding to each of those components within the dusty mixtures, that is, the dust coarse (Dc), $\beta_{Dc}$, dust fine (Df), $\beta_{Df}$, and non-dust (ND), $\beta_{ND}$. Note that $\beta_p = \beta_{Dc} + \beta_{Df} + \beta_{ND}$ ($z$-dependence is omitted for simplicity). Particular pure depolarization ratios of 0.39, 0.16 and 0.05 were assumed, respectively, for the Dc, Df, and ND components (Córdoba-Jabonero et al., 2018; Ansmann et al., 2019). Hence, the vertical PEC profile for each separated component can be obtained by definition as

$$\sigma_i(z) = S_a^i \times \beta_i(z), \tag{5}$$

where $i$ refers to the Dc, Df, and ND components, being $S_a^i$ their corresponding pure LR, which are assumed to be 55 sr for Dc and Df, and 25 sr for ND components (Ansmann et al., 2019). The vertical total PEC, $\sigma_p(z)$, can be calculated as follows

$$\sigma_p(z) = \sigma_{Dc}(z) + \sigma_{Df}(z) + \sigma_{ND}(z), \tag{6}$$

being

$$\tau = \sum_{n=1}^{n=N} \sigma_p(z_n)\, \Delta z \tag{7}$$

the total aerosol extinction in the overall atmospheric column (i.e., the AOD), where $\Delta z$ is the lidar vertical resolution, and $n$ indicates the discrete n-bin height-level ($n = 1, \ldots, N$), being $z_N$ the reference height under aerosol-free conditions.

## 3   Results

### 3.1 Experimental overlap function $F_{OVP}$

P-MPL observations were carried out from 6 June to 26 July 2019 at the Leipzig station during the field campaign. Simultaneous P-MPL and Polly/MARTHA measurements as performed under relatively clean conditions were selected for the OVP calibration purpose. The first comparison analysis corresponded to 12 hourly-averaged P-MPL and Polly RCS profiles within the night-time period from 28 June 2019 at 18UT to 29 June 2019 at 05UT (day-time values on 28 June at 18UT: AOD=0.10, Ångstrom exponent AE=1.59). The second one was related to the MARTHA night-time RCS measurements as averaged for 4 hours from 23 July 2019 at 21UT to 24 July 2019 at 00UT (day-time values on 23 July at 18UT: AOD=0.09, AE=1.33); P-MPL RCS profiles were also averaged during that same period for comparison. **Figure 1** shows the uncorrected by overlap P-MPL RCS profiles in comparison with the reference Polly (left panel) and MARTHA (right panel) ones for both those particular periods. The part of the P-MPL RCS profiling to be OVP-corrected is clearly highlighted ranging from the surface up to around 6 km height. Next, the experimental estimation of $F_{OVP}$ for the P-MPL system is analysed in terms of the OVP-corrected RCS as obtained by applying each of those experimentally-estimated $F_{OVP}^{Polly}$ and $F_{OVP}^{MARTHA}$ (see **Sect. 2.3.1**), including also a comparison with the original one, $F_{OVP}^{original}$ (as provided by the manufacturer).

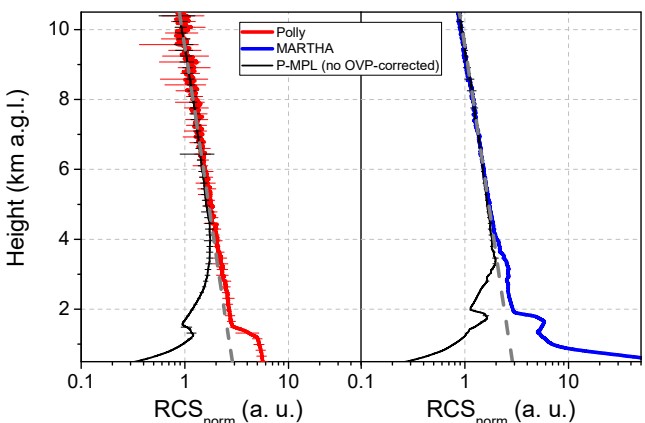

**Figure 1: Comparison of the (left) reference Polly (red line; for clarity, the 12 P-MPL and Polly RCS profiles, from 28 June 18UT to 29 June 05UT, were averaged) and (right) MARTHA (blue line; 4 P-MPL and MARTHA RCS profiles, from 23 July 21UT to 24 July 00UT, were averaged) w.r.t. the uncorrected by overlap P-MPL profiles (black lines). Normalization height at 9.5 km a.g.l. The aerosol-free background signal is shown by a grey dashed line.**


**Figure 2** shows the experimental OVP functions, $F_{OVP}(z)$, as obtained from the comparison of the P-MPL RCS profiles w.r.t. Polly and MARTHA lidar measurements (top panel, $F_{OVP}^{Polly}$ in red, and $F_{OVP}^{MARTHA}$ in blue) (see **Eq. 2**) together with $F_{OVP}^{original}$; associated errors are also shown in the bottom panel. In addition, as both those OVP functions were obtained in two different days, temperature-related changes

could be produced in the OVP calibration. Hence, the averaged $F_{OVP}^{av}(z)$ between both OVP functions is also calculated, and shown together the absolute and relative errors in **Fig. 2**, top and bottom panels, respectively). Details on the OVP error processing are described in **Annex A**. By comparing with the original OVP function, large discrepancies can be clearly observed, highlighting the change of $F_{OVP}(z)$ with time, mostly in the relevant 1-5 km height-range. Regarding the OVP functions $F_{OVP}^{Polly}$ and $F_{OVP}^{Marth}$ ,

differences are also found, mostly in the near-field range up to around 3 km height. However, by using $F_{OVP}^{av}(z)$ instead of one of two others for P-MPL RCS correction, its relative error is just $14 \pm 5\%$ in average from 0.3 up to 10 km height (see **Fig. 2-bottom**). Taking into account these errors, $F_{OVP}^{av}(z)$ can be the calibration function used for correcting the P-MPL RCS profiles at near-field heights, following the expression in **Eq. 1**, as it seems to be the best proxy for OVP correction of the P-MPL RCS profiles.

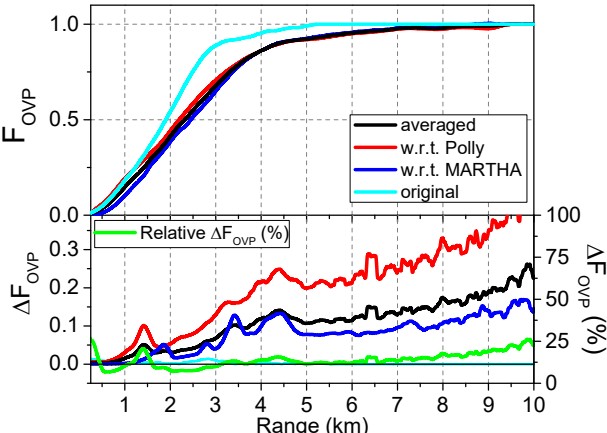


**Figure 2: (Top)** Experimental overlap functions, $F_{OVP}$, as obtained for two different days from the ratio between the P-MPL RCS profiles w.r.t. the Polly ($F_{OVP}^{Polly}$, red) and MARTHA ($F_{OVP}^{MARTHA}$, blue) ones, together with the averaged function ($F_{OVP}^{av}$) of both of them (black line); the original overlap function as provided by the manufacturer, $F_{OVP}^{original}$, is also included (cyan line). **(Bottom)** Errors, $\Delta F_{OVP}$, associated to the OVP-function

estimation for each comparison case: P-MPL w.r.t. Polly (red), P-MPL w.r.t. MARTHA (blue), and the averaged OVP function of both of them (black); the error for $F_{OVP}^{original}$ (cyan) and the relative error for $F_{OVP}^{av}$ (green line) are also included.

The previous uncorrected and OVP-corrected P-MPL RCS profiles by using both $F_{OVP}^{av}$ and $F_{OVP}^{original}$ are

shown in **Figure 3**. Slightly differences are observed for the P-MPL RCS profiles as compared to those Polly and MARTHA ones by using $F_{OVP}^{av}$, despite it was calculated from averaging $F_{OVP}^{Polly}$ and $F_{OVP}^{MARTHA}$, which were obtained from measurements on different days (only almost one month between them). Large differences are clearly found when $F_{OVP}^{original}$ is applied, mostly between 1.5 and 3 km height, evidencing that the OVP function as provided by the manufacturer is not applicable after some time for aerosol

research, being necessary an regular OVP recalibration, as performed and described in this work. Once the P-MPL RCS profiles are OVP-corrected, the optical properties of the aerosols can be retrieved using inversion algorithms. OVP-induced effects in the inversion of the aerosol optical properties are analysed in **Sect. 3.3**.



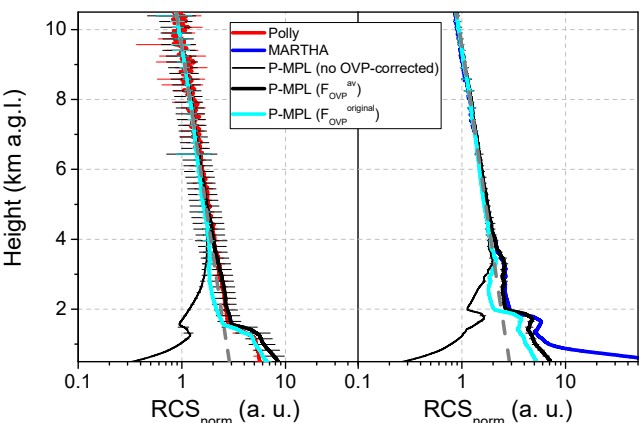

**Figure 3: OVP-corrected (black thick lines) P-MPL RCS profiles by using $F_{OVP}^{av}$ function and the uncorrected RCS ones (black thin lines), w.r.t. (Left) Polly (red line) and (Right) MARTHA (blue line) RCS profiles, together with the OVP-corrected ones by $F_{OVP}^{original}$ (cyan lines).**

**3.2 Volume linear depolarization ratio (VLDR)**

Before analysing the OVP impact in the retrieval of the aerosol optical properties, the VLDR is also examined. Despite the VLDR is unaffected by the OVP calibration, it actually affects, together with the PBC, $\beta_p$, the PLDR, $\delta_p$, estimation (see **Sect. 2.3.2**).

The P-MPL VLDR is calculated using **Eq. 8** and compared with that derived from Polly measurements as reference, since TROPOS follows all quality assurance efforts regarding polarization lidar calibration
tests in the Polly systems as recommended by EARLINET (Freundenthaler et al., 2008, 2016). A dust case observed at Leipzig site for the night on 29-30 June 2019 is examined for that purpose. **Figure 4** shows the VLDR as obtained from both the $\delta_{MPL}^V$ and $\delta_{Polly}^V$ profiles as averaged from 18 to 23 UT on 29 June and from 00 to 05 UT on 30 June (for clarity, only averaged $\delta^V$ profiles are shown). The dust signature is clearly marked, showing a dust layer clearly confined between 3 and 6 km height, with a
higher variability for the second interval due to the decay of dusty conditions at the end of that period, as reflected by a larger error uncertainty in time averaging. In overall, $\delta_{MPL}^V$ values are higher than those $\delta_{Polly}^V$, peaking between 0.11 and 0.14 within the dust layer. Hence, the VLDR was averaged within several aerosol-free height-intervals, below and above that defined dust layer, to analyse potential changes and offsets. Those mean $\delta^V$ values (and their standard deviation, SD) are shown in **Table 2**.





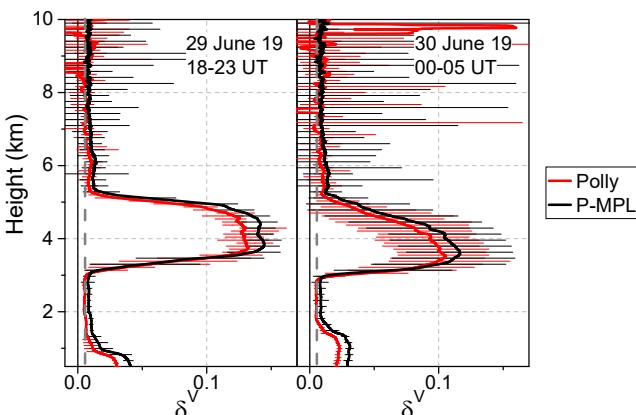


**Figure 4: Volume linear depolarization ratio (VLDR), $\delta^V$, as obtained from both the P-MPL (black line) and Polly (15-p smoothed red line) measurements carried out on: (left) 29 June 2019, as averaged from 18 to 23 UT, and (right) 30 June 2019, as averaged from 00 to 05 UT (error bars are also shown in black and red, respectively). The aerosol-free background $\delta^V$ is marked by a grey dashed line.**


**Table 2: Mean VLDR values together their standard deviation (SD) (and their relative SD error, in %) as obtained from the P-MPL and Polly measurements ($\delta_V^{MPL}$ and $\delta_V^{Polly}$ profiles) for aerosol-free height-intervals on 29-30 June 2019.**

| Height intervals | $\delta^V$, mean ± SD (%SD) | |
|---|---|---|
| (km) | P-MPL | Polly |
| 1.5-2.5 | 0.0096 ± 0.0016 (16.6) | 0.0057 ± 0.0002 (3.4) |
| 7.0-8.0 | 0.0088 ± 0.0010 (10.8) | 0.0057 ± 0.0037 (65.9) |
| 8.0-9.0 | 0.0083 ± 0.0016 (19.7) | 0.003 ± 0.016 (> 100) |
| Height-averaged | 0.0089 ± 0.0005 (6.0) | 0.0049 ± 0.0011 (23.1) |

Looking at the results, $\delta_{Polly}^V$ presents larger errors than those for $\delta_{MPL}^V$, as associated to a lower signal-to-noise ratio as height increases for the Polly measurements (no smoothing applied). This is reflected by the higher relative error (%SD) found for the Polly VLDR (23%) w.r.t. to that for the P-MPL (6%) when all the aerosol-free height-intervals are considered, being the mean $\delta^V$ values of 0.0089 ± 0.0005 (%SD: 6%) and 0.0049 ± 0.0011 (%SD: 23%), respectively, for the P-MPL and Polly VLDR. As a result, an constant

offset, $\Delta$ (= $\delta_{Polly}^V - \delta_{MPL}^V$), can be assumed between $\delta_{MPL}^V$ and $\delta_{Polly}^V$, obtaining $\Delta$ = -0.0040 ± 0.0016. This offset represents a correction to account for any slight mismatch in the transmitter and detector polarization planes and any impurity of the laser polarization state (Sassen, 2005), as also found in Córdoba-Jabonero et al. (2013) by characterizing the VLDR of a relatively older version (MPL-4) of the polarized MPL systems. Therefore, the P-MPL VLDR must be also corrected by that offset using the

expression:

$$\delta_{MPL}^{V\,corr} = \delta_{MPL}^V + \Delta, \qquad\qquad\qquad (8)$$



where $\delta_{MPL}^{V\,corr}$ is the corrected P-MPL VLDR profile.

Regarding the dust layer extended between 3.5 and 5.0 km height, as expected, a similar $\delta^V$ value to that obtained for the Polly VLDR ($\delta_{Polly}^V = 0.11 \pm 0.02$) is estimated for the corrected P-MPL VLDR, i.e.,

$\delta_{MPL}^{V\,corr} = 0.12 \pm 0.02$, as averaged within that dust layer. The corresponding PLDR to those $\delta^V$ are around 0.3 (as shown in **Sect. 3.3**), which are typical PLDR values for dust (Burton et al., 2012; Gross et al., 2013).

### 3.3 Particle backscatter coefficient (PBC) and particle linear depolarization ratio (PLDR)

The effect of the OVP correction on the P-MPL RCS is also analysed regarding the retrieval of the KF-

derived $\beta_p$ profiles, as obtained by applying both $F_{OVP}^{original}$ and $F_{OVP}^{av}$ to the RCS. A dust event as observed at Leipzig on the night from 29 to 30 June 2019 (the same dust case as previously exposed in **Sect. 3.2**) is selected for that purpose. In addition, both PLDR, $\delta_p$ (see **Eq. 3**), and VLDR, $\delta^V$ (see **Eqs. 4 and 9**, $\Delta$ offset corrected) are estimated. The OVP-induced effect is illustrated, in particular, using the vertical hourly-averaged profiling observed on 29 June 2019 at 20-21 UT, corresponding to a well-separated two-

layer dust case (dust optical depth of 0.061). **Figures 5 and 6** show the vertical profiles of $\beta_p$ and $\delta_p$ (and $\delta^V$), respectively, depending on the $F_{OVP}$ applied, as retrieved from the P-MPL measurements together to those derived from Polly ones for the selected case.

Both P-MPL and Polly datasets show a dust layer clearly confined between around 3.5 and 5.0 km height. For comparison, in addition to the AOD-constrained KF solution for the PBC (reference height at 6.0 km,

and reference backscatter coefficient of $10^{-7}$ Mm$^{-1}$ sr$^{-1}$) using $S_a^{eff} = 43$ sr (that obtained from Polly elastic measurements) (see **Figs. 5a and 5c**), $\beta_p$ is also retrieved by using the Raman-derived LR ($S_a^{Raman} = 60$ sr) for that dust layer as obtained from the night-time Polly Raman measurements (data not shown) (see **Figs. 5b and 5d**).

Table 3: Dust layer-averaged PBC, $\overline{\beta_p}$ (Mm$^{-1}$ sr$^{-1}$), and PLDR, $\overline{\delta_p}$, and the integrated backscatter, $B$ ($10^{-3}$ sr$^{-1}$), values, as obtained from P-MPL $\beta_p$ and $\delta_p$ profiles on 29 June 2019 at 20-21 UT in dependence of the $F_{OVP}$ applied for both the KF solutions (using $S_a^{eff}$ and $S_a^{Raman}$). Corresponding Polly values are also included.

| | P-MPL | | | | | | Polly | | |
|---|---|---|---|---|---|---|---|---|---|
| $F_{OVP}$ | $S_a^{eff} = 43$ sr | | | $S_a^{Raman} = 60$ sr | | | $S_a^{eff} = 43$ sr | | |
| | $\overline{\beta_p}$ | $B$ | $\overline{\delta_p}$ | $\overline{\beta_p}$ | $B$ | $\overline{\delta_p}$ | $\overline{\beta_p}$ | $B$ | $\overline{\delta_p}$ |
| $F_{OVP}^{av}$ | 0.93 ± 0.17 | 1.41 ± 0.16 | 0.32 ± 0.01 | 0.89 ± 0.15 | 1.35 ± 0.16 | 0.33 ± 0.01 | | | |
| $F_{OVP}^{Polly}$ | 0.92 ± 0.16 | 1.40 ± 0.27 | 0.32 ± 0.01 | 0.88 ± 0.14 | 1.33 ± 0.27 | 0.33 ± 0.01 | 0.72 ± 0.16 | 1.08 | 0.29 ± 0.03 |
| $F_{OVP}^{MARTHA}$ | 0.94 ± 0.17 | 1.43 ± 0.10 | 0.32 ± 0.01 | 0.90 ± 0.15 | 1.36 ± 0.10 | 0.32 ± 0.01 | | | |
| $F_{OVP}^{original}$ | 0.87 ± 0.14 | 1.32 ± 0.05 | 0.33 ± 0.01 | 0.83 ± 0.12 | 1.26 ± 0.08 | 0.34 ± 0.02 | | | |





Regarding the dust layer, relatively small differences are found between Polly and P-MPL $\beta_p$ profiles

(see **Figs. 5a** and **5b**), at least within error uncertainties. In order to assess those differences between both

datasets, the layer-averaged PBC, $\overline{\beta_p}$ (Mm$^{-1}$ sr$^{-1}$), and the integrated backscatter, $B$ (sr$^{-1}$), for this 3.5-5.0-

km dust layer were calculated to be used as a proxy of the degree of agreement. Derived $\overline{\beta_p}$ and $B$ values

in dependence of $F_{OVP}$ for both the KF solutions (using either $S_a^{eff}$ or $S_a^{Raman}$) are shown in **Table 3**. In

general, $\overline{\beta_p}$ and $B$ are higher for P-MPL w.r.t. Polly retrievals. Concerning the KF solutions for P-MPL

profiles, a better agreement is achieved when the $S_a^{Raman}$ of 60 sr is applied (no AOD-constrain), i.e.,

lower differences for $\overline{\beta_p}$ and $B$ are found w.r.t. Polly-retrieved values.

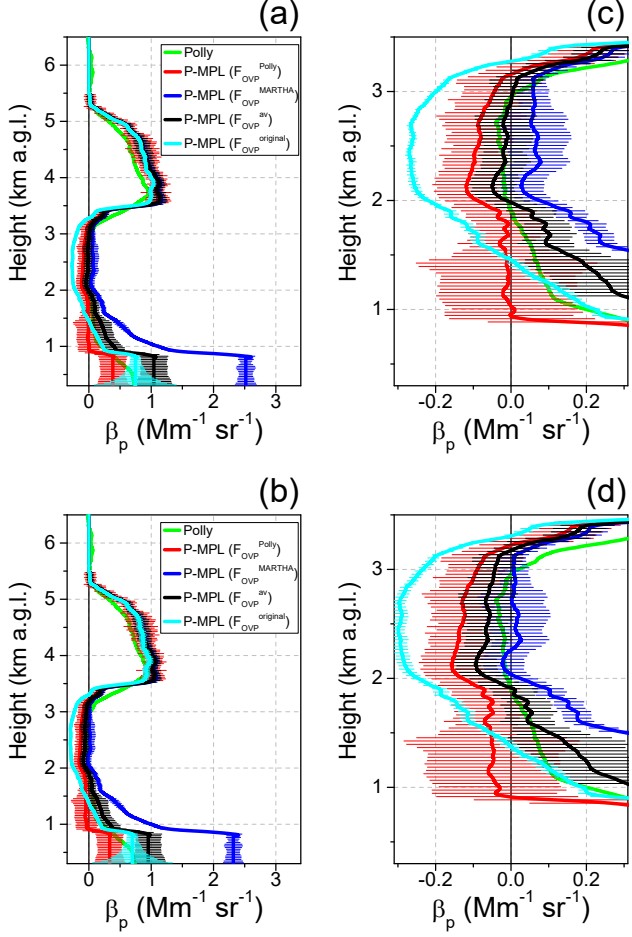

**Figure 5: Dust case as observed on 29 June 2019 at 20:00-21:00 UT over Leipzig: Vertical particle backscatter**

**coefficient (PBC), $\beta_p$, as retrieved in dependence of the OVP function applied to the P-MPL RCS: $F_{OVP}$ w.r.t.**

**to Polly (red) and MARTHA (blue) data and both the $F_{OVP}^{av}$ (black) and $F_{OVP}^{original}$ (cyan) by using the KF**

**solution with (a) the elastic AOD-constrained LR ($S_a^{eff}$ = 43 sr), and (b) the Raman-retrieved LR ($S_a^{Raman}$ = 60**

**sr) for the dust layer; and (c) and (d) the same as Fig. 5-a and 5-b, respectively, but highlighting the near-field**



range of $\beta_p$ between 0.5 and 3.5 km height (the x-axis is also accordingly scaled). Corresponding Polly-

retrieved $\beta_p$ profiles are also included (green lines).

Nevertheless, the KF retrieval is mostly affected at near-field ranges (up to 3 km height) (see **Figs. 5b** and

**5d**), as expected, since the OVP correction is rather relevant at those ranges. Negative $\beta_p$ values are

predominantly found for the scenarios when the RCS is OVP-corrected by $F_{OVP}^{Polly}$ and $F_{OVP}^{original}$, being

more pronounced when the $S_a^{Raman}$ is applied, since the LR to be applied in this height-interval must be

closer to the elastic $S_a^{eff}$ of 43 sr. The best fitting seems to be achieved by using $F_{OVP}^{MARTHA}$ and $F_{OVP}^{av}$.

Among those, however, results show that $\beta_p$ profiles are in a better agreement by using $F_{OVP}^{av}$ as compared

to those Polly-derived $\beta_p$ at ranges from around 1 km down (see **Figs. 5a** and **5c**). Relative $\beta_p[F_{OVP}^{av}]$

errors of 10-20% are obtained.

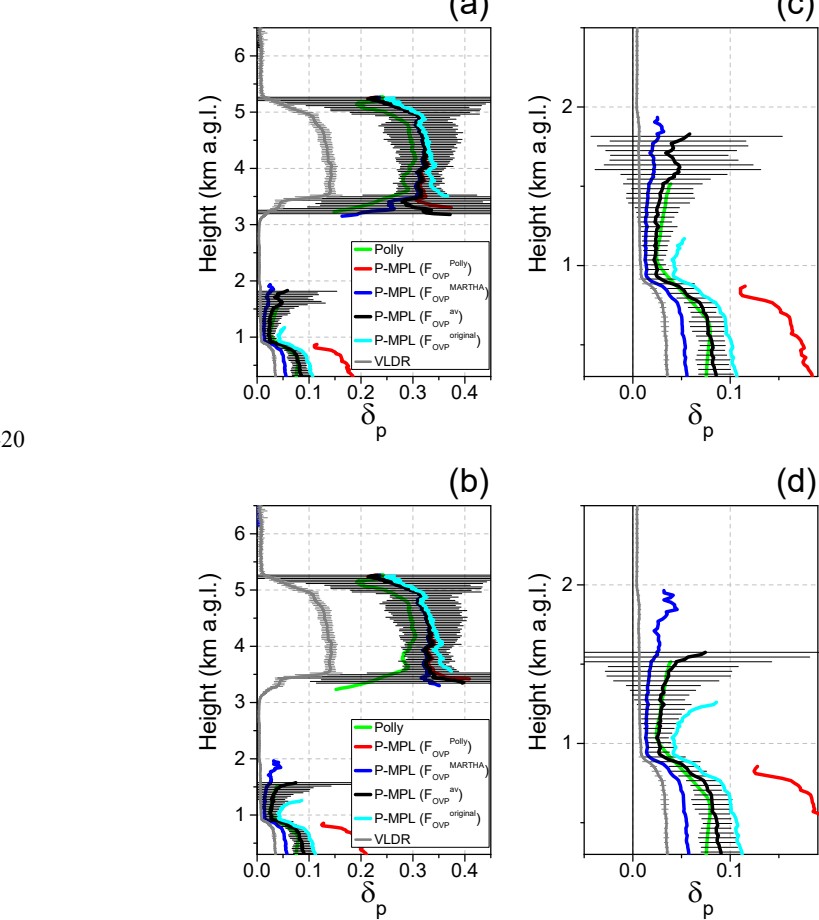


**Figure 6: The same as Fig. 5, but for the vertical particle linear depolarization ratio (PLDR), $\delta_p$, as retrieved**

**from each $\beta_p[F_{OVP}]$ as shown in Fig. 5, and the VLDR, $\delta^V$ (grey line). The corresponding Polly-retrieved $\delta_p$**





profile is also included (green line). For clarity, only error bars are marked for $\delta_p[F_{OVP}^{av}]$ (black) and $\delta_p^{Polly}$
(green).

By examining the PLDR profiles, the dust signature is also clearly marked between around 3.5 and 5.0 km height, i.e., typical $\delta_p$ values for dust of around 0.3 are found (see **Table 3**), indicating a predominance of coarse particles. No large differences are found between Polly and P-MPL PLDR
profiles for that layer (see **Figs. 6a** and **6b**), with mean $\delta_p$ values of 0.29 (Polly) and 0.31-0.34 (P-MPL, depending on the $F_{OVP}$ applied and the LR used) (see **Table 3**).

### 3.4  Particle extinction coefficient (PEC) retrieval for a dust case study

Once $F_{OVP}^{av}$ is experimentally determined to correct the P-MPL RCS profiles (see **Sect. 3.1**), the total PBC can be retrieved by using the KF algorithm (a $S_a^{eff}$ is also estimated against AERONET AOD constraint).
The PLDR was also obtained by using the PBC and the corrected VLDR (see **Sect. 3.2**). In addition, using the approach as described in **Sect. 2.4.2**, the vertical total PEC profile, $\sigma_p$, and that corresponding to each component, $\sigma_i$ ($i$ = Dc, Df, ND), are obtained (see **Eqs. 5-7**), for instance, during a dusty event occurred at Leipzig in June 2019. For illustration, two different dust scenarios are examined: a well-separated pure dust layer observed on 29 June at 20-21 UT, and a mixed dust case occurred on 30 June at
16-17 UT.

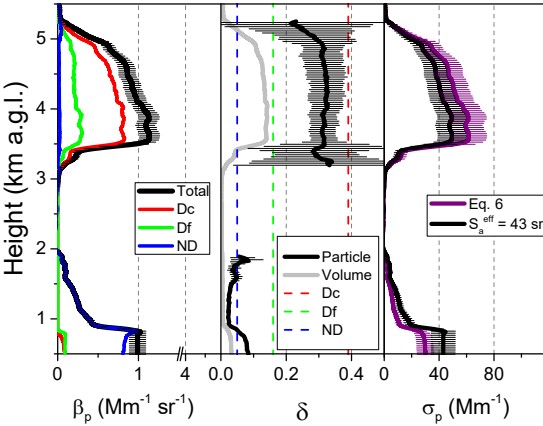

**Figure 7: Dust scenario: well-differentiated pure dust layer on 29 June 2019 at 20-21 UT. Profiles of the (Left) particle backscatter coefficient: total (black), Dc (red), Df (green) and ND (blue); (Centre) linear depolarization ratio: particle (black) and volume (grey) together with those assumed for the pure components;**
**and (Right) total particle extinction coefficient as obtained from: the use of an AERONET-constrained effective LR (black) and the sum of the particular PEC for each component (Dc, Df and ND) (Eq. 6, purple).**

Figures 7 and 8 show the vertical profiles of both the total backscatter (and also that detached into the Dc, Df and ND components) and extinction coefficient together with the PLDR and VLDR, respectively,
for each of those two selected cases. Differences in the vertical structure of the $\sigma_p$ profiles between those





by using the KF-derived effective LR ($S_a^{eff}$ = 43 and 25 sr are obtained, respectively, for the pure and mixed dust cases) and those by applying **Eq. (6)** are clearly shown for both dust scenarios.

In general, the KF solution for the total extinction seems to be underestimated, although $\tau$ values as obtained from **Eq. 7** differ from the effective total extinction (by using $S_a^{eff}$) in -6.4% and +25.7%,

respectively, for the pure ($\tau$ = 0.103; **Fig.7**) and mixed ($\tau$ = 0.264; **Fig. 8**) dust cases. However, in those particular dust scenarios, the largest discrepancies are mostly found in layers with dust (Dc and Df) detection (see $\beta_{Dc}$ and $\beta_{Df}$ profiles, i.e., red and green lines, respectively, in **Figs. 7-left and 8-left**), regarding the difference in the LR applied between the effective value and that assumed typical for dust (55 sr). This is mainly observed for the dust mixed case (**Fig. 8**), as that LR difference is higher than that

for the pure case. Indeed, the total extinction associated to layers with dust predominance is +26.7% and +56.7% of the corresponding effective value found, respectively, for the pure (3.5-5.0 km; see **Fig. 7-right**) and mixed (1.5-5.0 km; see **Fig. 8-right**) dust layers.

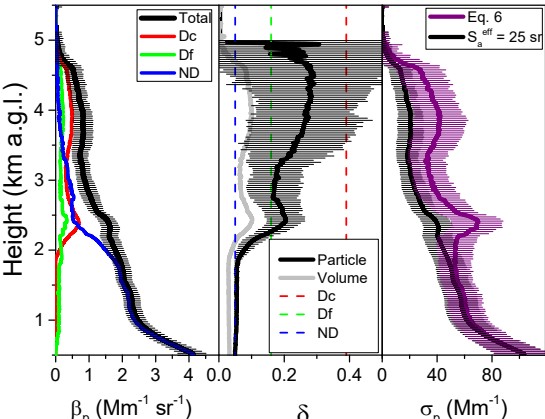

**Figure 8: The same as Fig. 7, but for the dust scenario: mixed dust case on 30 June 2019 at 16-17 UT.**


Therefore, the crucial point is concerning to the particular vertical aerosol extinction layering that is estimated by using either the effective KF-derived solution or the introduced depolarization-based method as observed in both dust scenarios. Indeed, this is especially relevant for the aerosol impact in atmospheric and climate research (atmospheric composition, radiative effect, cloud nucleation, …).

Moreover, this method can be easily used as an alternative approach for extinction retrieval in other elastic polarized lidar systems.

## 4    Conclusions

A comprehensive two-month field campaign has been performed in summer 2019 to characterize the performance of a polarized Micro-Pulse Lidar (P-MPL) system, and to check the quality of the retrieved

products. Atmospheric observations with the P-MPL system, currently operative within MPLNET, have been examined against those from two referenced EARLINET lidars (Polly and MARTHA), which are



operative at Leipzig site (Germany, 51.4ºN 12.4ºE, 125 m a.s.l.) as managed by TROPOS. In particular, a characterization assessment in terms of the overlap (OVP) correction and its impact in the retrieval of the optical properties has been achieved. Furthermore, the volume linear depolarization ratio (VLDR) has also been cross-checked and corrections applied, allowing an accurate retrieval. The aim of this work has been focused on the determination of the lidar-specific true OVP function and on investigating in detail the accuracy of both the retrieved particle backscatter coefficient (PBC) and particle linear depolarization ratio (PLDR) profiles.

It has been highlighted that the OVP function as delivered by the P-MPL manufacturer cannot be used. The reasons are manifold, but an experimental assessment of the OVP calibration should be recommend for the MPL systems. The experimental procedure to determine the OVP function for the P-MPL system has been described in the basis of the comparison to reference lidars. The optimal OVP function for correcting our P-MPL measurements has been experimentally obtained, together with its uncertainties, under clean observational conditions from simultaneous P-MPL and Polly/MARTHA observations, and compared with the original one as provided by the manufacturer. In addition, depending on the OVP function applied, the calibration-induced effects on the retrieval of both the PBC and PLDR for the P-MPL system have been analysed for two KF solutions by using either the elastic (AOD-constrained) or the Raman-provided lidar ratios in comparison with those PBC and PLDR retrievals as obtained from simultaneous Polly observations.

Additionally, the VLDR has been also examined in comparison with the Polly VLDR regarding its effect in the PLDR determination. A suitable VLDR profile has been usually obtained, being only needed to be corrected by a small offset value, which has been also estimated.

Once P-MPL measurements were optimally OVP-corrected, the PBC, and also the PLDR, profiles have been accurately derived by using the KF solution (an effective LR is obtained in constraint with AERONET AOD). In addition, an alternative method has been introduced to derive the vertical particle extinction coefficient (PEC) profiles from elastic P-MPL measurements in combination with the POLIPHON algorithm by separating the optical properties into those corresponding to each component within aerosol mixtures. A dust event occurred at Leipzig in June 2019 is used for illustration, selecting two different dust scenarios: a well-differentiated dust layer and a mixed dust case. This is especially relevant for elastic lidars, as the P-MPL system among others, due to the indetermination in solving the lidar equation.

In overall, an adequate OVP function is needed to be determined in a regular basis in order to calibrate the P-MPL system and, hence, to derive suitable aerosol products (backscatter, depolarization, extinction).

**Annex A**

The experimental overlap (OVP) function, $F_{OVP}^{ref}(z)$, is obtained from the expression

$$F_{OVP}^{ref}(z) = P^{MPL}(z)/P^{ref}(z), \tag{A.1}$$


where $P^{MPL}(z)$ are the P-MPL RCS profiles, which are compared against those reference lidar measurements, $P^{ref}(z)$ ($ref$ denotes either Polly or MARTHA) using the experimental approach as described in this work.

The error associated to the determination of the OVP function, $\Delta F_{OVP}$, is obtained from error propagation calculations of the **Eq. A.1**. In this sense, it can be expressed as ($z$-dependence is omitted for simplicity, hereafter)

$$\Delta F_{OVP}^{ref} = F_{OVP}^{ref} \times \left[ \frac{\Delta P^{MPL}}{P^{MPL}} + \frac{\Delta P^{ref}}{P^{ref}} \right],$$

(A.2)

where $\Delta P^{MPL}$ and $\Delta P^{ref}$ are, respectively, the errors related to $P^{MPL}$ and $P^{ref}$.

$\Delta P^{MPL}$ can be estimated as composed of two error contributions: one associated to instrumental corrections (energy fluctuations, instrumental calibrations, solar background, …), $\varepsilon^{MPL}$, as described in Welton and Campbell (2002), and another one reflecting the atmospheric variability within the time-averaging performed of the $P^{MPL}$ profiles, which is expressed by the standard deviation, $sd^{MPL}$; hence, it can be obtained from the expression

$$\Delta P^{MPL} = \sqrt{(\varepsilon^{MPL})^2 + (sd^{MPL})^2}.$$

(A.3)

Errors associated to the reference lidar measurements, $\Delta P^{ref}$ ($ref$ is for either Polly or MARTHA), are represented by the standard deviation, as obtained from the corresponding time-averaging of $P^{ref}$ profiles.

In this work, the averaged function between $F_{OVP}^{Polly}$ and $F_{OVP}^{MARTHA}$ is also calculated, i.e.,

$$F_{OVP}^{av} = \frac{F_{OVP}^{Polly} + F_{OVP}^{MARTHA}}{2},$$

(A.4)

being the error related to this function, $\Delta F_{OVP}^{av}$, estimated as

$$\Delta F_{OVP}^{av} = \sqrt{\left( \frac{\Delta F_{OVP}^{Polly}}{2} \right)^2 + \left( \frac{\Delta F_{OVP}^{MARTHA}}{2} \right)^2},$$

(A.5)

where $\Delta F_{OVP}^{ref}$ ($ref$ denotes either Polly or MARTHA) is the error as obtained from **Eq. A.2**.

**Data availability.** All data generated and analysed for this study are available from the authors upon reasonable request.

**Author Contributions.** CC-J and AA designed the study and wrote the original draft paper. CC-J, AA, CJ and HB provided data. CC-J and CJ performed data analysis with contributions from AA, HB, MAL-C and RE. All authors reviewed and edited the final version of the manuscript. All the authors agreed to the final version of the paper.

**Competing interests.** The authors declare that they have no conflict of interest.



**Acknowledgements**

This work was supported by the Spanish Ministry of Science, Innovation and Universities (MCIU) under grants PRX18/00137 (Programa "Salvador de Madariaga") and CGL2017-90884-REDT (ACTRIS-Spain), the Spanish Ministry of Science and Innovation (MICINN) (grant PID2019-104205GB-C21), and the H2020 program from the European Union (ACTRIS, GA n. 778349). MALC is supported by the INTA training fellowship programme. The MPLNET project is funded by the NASA Radiation Sciences Program and Earth Observing System.

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
