# Peer review of "Experimental assessment of a Micro-Pulse Lidar system in comparison with reference lidar measurements for aerosol optical properties retrieval"

_Atmospheric Measurement Techniques, 2020_

## Referee Comment (RC1) · Anonymous Referee #1 · 26 Nov 2020

Authors compare MPL data with measurements of Raman lidars, to evaluate the overlap function and estimate it's influence on backscattering coefficient and depolarization ratio. This is useful technical study, which, by my opinion, can be published in AMT after minor revision.

I have just technical comments

Ln.143. "Those two polarized signals are semi-simultaneously detected by alternatively switching in the basis of 50%/50% the LRC polarization mode (LCR switching time of

133 $\mu$s) within every integrating minute." Unclear. Switching occurs every minute or every pulse?

Ln.259. "and 25 sr for ND components". Why so small value? For example, for smoke it can be 70 sr.

Eq.5,6. I am confused. To calculate extinction profile assumptions about lidar ratios for all three components are made. Is it still more accurate than just apply Klett solution?

Ln.333. "The P-MPL VLDR is calculated using Eq. 8" I don't see Eq.8.

Ln.364. "Therefore, the P-MPL VLDR must be also corrected by that offset using..." But in calculation of VDR from Polly data, the calibration coefficient is used. Can corresponding uncertainty contribute to this offset?

Ln 377. "see Eqs. 4 and 9..." I don't see Eq.9.

Fig.5. I didn't understand what is difference between (a,b) and (c,d). Are (c,d) plots necessary? The same about Fig.6.

Fig.8. I don't quite understand why authors decompose extinction for three components. Looks like goal of the paper is to correct the overlap function.

---

## Referee Comment (RC2) · Anonymous Referee #2 · 22 Jan 2021

The manuscript fits within the journal scope, as it describes the results from an inter-comparison campaign in order to evaluate the Micro Pulse lidar overlap function taking EARLINET Martha and Polly systems as reference.

The manuscript is interesting, nevertheless some major changes are needed before publication.

1) I understand that it is not very practical to find an horizontal line of sight free from obstacles with an homogenous atmosphere, but I think that this setup is way easier

than organizing a measurement campaign on purpose. Moreover, shooting the lidar horizontally is more accurate than the proposed method.

2) The manuscript needs a deep English editing, because some parts are not clear at all. I was editing some parts, but it is not a reviewer role.

3) Some sections in the manuscript seem to be out of context. As stated in the title and mostly in the abstract, the main objective is to calibrate the MPL instruments with respect to the reference EARLINET lidars. The part where POLIPHON algorithm is applied is not adding value to the paper with respect to its main goal. I suggest to the authors to better contextualize it (maybe editing English will make it clearer) or delete it. Moreover, I think that the retrieval doesn't make so much sense. First, Leipzig is not the best spot to detect dust outbreaks, because the aerosol layer traveled so much before reaching the observation site. Then dividing the backscattering coefficient into those 3 categories is rather audacious and potentially wrong. There is not any information regarding the aerosol size distribution. Then Dc and Df how are assessed ? Just using the Particle Depol Ratio and the LR? In this case, no information is available on how the dust particles aged, i.e. if dust mixes up with urban or continental aerosol. Also, the used values are probably found for some specific measurement campaigns and cannot be assumed valid in general. For this reason, those values will show a very high variability making the error on retrieval huge. What if, during the advection, the dust particles mix with other aerosol particles? The LR changes, the depolarization changes...

4) Being the P-MPL a product commercially available, it is not possible to establish with precision which technology is used to detect the depolarized laser light, because, as stated on MPLNET website, there exist at least two different P-MPL models that depend on fabrication year. For the P-MPL models produced before 2013, the use of nematic liquid crystal polarizer introduces a delay in data rates. A new P-MPL model was developed around 2013 following Flynn et al 2007, but using a ferroelectric liquid crystal (FLC) for faster data rates and a slightly modified measurement strategy to

accommodate the difference in polarizer properties. For this reason, as long as a proper instrument characterization and stability study of the polarized design and its calibration procedures will be not available, equation 4 and section 3.2 are based on speculations.

Specific comments are found in the attached files.

Please also note the supplement to this comment:
https://amt.copernicus.org/preprints/amt-2020-427/amt-2020-427-RC2-supplement.pdf

---

## Author Comment (AC1) · 2 Apr 2021

**Authors' response (in blue) to the Reviewer #1's comments (in black):**

The authors thank Reviewer #1 for their comments and suggestions that definitely improved the manuscript. Required changes and modifications have been introduced in the text of the revised version of the manuscript by using the Word Track Changes tools.

In general, the title has been modified, and following some reviewer #2's suggestions, Sections 2.4.2 and 3.4 have been removed and the proposed changes as indicated in the Supplement by the reviewer #2 have been implemented as well. New references have been added and Figures 5 and 6 have been simplified, as well.

Next, the authors respond to the particular comments of the reviewer #1.

**- Reviewer 1**

Authors compare MPL data with measurements of Raman lidars, to evaluate the overlap function and estimate it's influence on backscattering coefficient and depolarization ratio. This is useful technical study, which, by my opinion, can be published in AMT after minor revision.

I have just technical comments

**R1C1.** Ln.143. "Those two polarized signals are semi-simultaneously detected by alternatively switching in the basis of 50%/50% the LRC polarization mode (LCR switching time of 133 μs) within every integrating minute." Unclear. Switching occurs every minute or every pulse?

Authors' response: The MPL system switches the polarization state every 250 pulses (but just 249 pulses are collected since one of the pulses is discarded during the ~100 μS it takes to switch). Therefore, the sentence is conservatively right, but in order to avoid the confusion, the corresponding text has been modified in the revised version of the manuscript as follows:

**Page 5, lines 160-162**: "Those two polarized signals are semi-simultaneously detected by alternatively switching in the basis of 50%/50% the LRC polarization mode within every integrating minute. Note that the P-MPL pulse frequency is 2500 Hz, and the polarization state is switched every 250 pulses, but just 249 pulses are collected since one of the pulses is discarded during the LCR switching time (~100 μs)."

**R1C2.** Ln.259. "and 25 sr for ND components". Why so small value? For example, for smoke it can be 70 sr.

Authors' response: That's true. However, a minor contribution of non-dust (ND) aerosols under dusty conditions is expected in comparison with the predominance of dust particles. Besides, smoke particles were not identified for the selected cases as shown

in the manuscript. The choice of 25 sr is just a conservative low value, which is assumed for the lidar ratio of ND aerosols by considering their small contribution, mainly within the dust layers.

**R1C3.** Eq.5,6. I am confused. To calculate extinction profile assumptions about lidar ratios for all three components are made. Is it still more accurate than just apply Klett solution?

Authors' response: For elastic lidars, an a-priori particle lidar ratio must be assumed. By using the KF algorithm in constraint with an ancillary value of AOD (i.e., AERONET AOD), an effective lidar ratio is obtained, which is a height-constant parameter. Hence, the height-resolved extinction coefficient is just obtained by multiplying the height-resolved backscatter coefficient and that effective lidar ratio; the AOD is calculated by integrating the extinction coefficient in height. Actually, the lidar ratio is not constant with the height as it is dependent on the aerosol type detected along the atmosphere, and therefore the extinction profile can differ in dependence on the aerosol mixing state of the atmosphere. In this work, the total extinction profile is obtained by summing the separated extinction coefficient profiles of each of the particle components, which the specific lidar ratio is indeed accurately known for. Therefore, in our case, we consider this is an improved result in the retrieval of a 'more accurate' extinction coefficient profile from elastic P-MPL lidar measurements, where a vertical lidar ratio, to some extent, has been intrinsically applied.

However, in the revised version of the manuscript, Sections 2.4.2 and 3.4, regarding the determination of the extinction profile, have been removed, as suggested by the reviewer #2.

**R1C4.** Ln.333. "The P-MPL VLDR is calculated using Eq. 8" I don't see Eq.8.

Authors' response: Eq. 8 of the original manuscript (page 12, line 366) is the renumbered Eq. 7 in its revised version (page 13, line 401). However, for avoiding any confusion, the text has been modified in the revised version as follows:

**Page 14, line 405**: "… where $\delta_{MPL}^{V}{}^{corr}$ is the corrected P-MPL VLDR profile, and $\delta_{MPL}^{V}$ is that VLDR as obtained from Eq. 2".

**R1C5.** Ln.364. "Therefore, the P-MPL VLDR must be also corrected by that offset using. . ." But in calculation of VDR from Polly data, the calibration coefficient is used. Can corresponding uncertainty contribute to this offset?

Authors' response: The calibration parameters of the continuously operated Polly lidar are automatically checked on a daily basis, and the calibration parameters are stored as time series (over months) to identify biases and miss-alignment and do corrections and

improve alignment. Hence, we assume no bias (in the Polly depolarization ratio) when all calibration parameters show good performance of the lidar over days, weeks, and months, as is the case here.

**R1C6.** Ln 377. "see Eqs. 4 and 9. . ." I don't see Eq.9.

Authors' response: Right. It was a mistake. Eq. 9 is actually the Eq. 8 (renumbered Eq. 7). This has been corrected in the revised version of the manuscript (page 13, line 404).

**R1C7.** Fig.5. I didn't understand what is difference between (a,b) and (c,d). Are (c,d) plots necessary? The same about Fig.6.

Authors' response: Plots in (c) and (d) are a 'zoom' of those (a) and (b); maybe they give a redundancy of results. Then, we will leave only plots (a) and (b) of both Figs. 5 and 6 in the revised version of the manuscript.

**R1C8.** Fig.8. I don't quite understand why authors decompose extinction for three components. Looks like goal of the paper is to correct the overlap function.

Authors' response: Indeed, the main goal is to experimentally assess the P-MPL system, including the determination of an overlap function for the P-MPL and the evaluation of its volume linear depolarization ratio (VLDR), in addition to study the effect in the lidar-derived aerosol optical properties, like the particle backscatter coefficient (PBC) and the particle linear depolarization ratio (PLDR). However, we considered consistent to extent that study to the other interrelated variables as the height-resolved extinction coefficient. In particular for elastic lidars as the P-MPL, it is relevant to estimate the extinction profile. Therefore, an alternative methodology to derive the extinction was also introduced in this work, which is based on the firstly separated three components (particularly, in dust mixtures), and then the total extinction is basically obtained by summing them, as described in Sect. 2.4.2.

However, in the revised version of the manuscript, Sections 2.4.2 and 3.4, regarding the determination of the extinction profile, have been removed (see **R1C3**), as suggested by the reviewer #2.

---

## Author Comment (AC2) · 2 Apr 2021

**Authors' response (in blue) to the Reviewer #2's comments (in black):**

The authors thank Reviewer #2 for their comments and suggestions that definitely improved the manuscript. Required changes and modifications have been introduced in the text of the revised version of the manuscript by using the Word Track Changes tools.

In general, the title has been modified, and following some reviewer #2's suggestions, Sections 2.4.2 and 3.4 have been removed and the proposed changes as indicated in the Supplement by the reviewer #2 have been implemented as well. New references have been added and Figures 5 and 6 have been simplified, as well.

Next, the authors respond to the particular comments of the reviewer #2.

**- Reviewer 2**

The manuscript fits within the journal scope, as it describes the results from an intercomparison campaign in order to evaluate the Micro Pulse lidar overlap function taking EARLINET Martha and Polly systems as reference.

The manuscript is interesting, nevertheless some major changes are needed before publication.

**R2C1.** I understand that it is not very practical to find an horizontal line of sight free from obstacles with an homogenous atmosphere, but I think that this setup is way easier than organizing a measurement campaign on purpose. Moreover, shooting the lidar horizontally is more accurate than the proposed method.

Authors' response: The main goal of the campaign, indeed, was not the determination of the overlap (OVP) function for the P-MPL system; but, taking this advantage, part of the observational period during the campaign was devoted to obtain an accurate OVP correction for that P-MPL system. Although shooting the lidar horizontally seems to be a more accurate setup than other methods for OVP determination, it cannot be always viable under regular conditions, at least neither at El Arenosillo (Huelva) station, where our P-MPL is routinely in operation, and nor at Leipzig. Our experience tell us that horizontal pointing of the lidar to obtain the overlap profile is not an easy and simple approach. It sounds simple, but it isn't in many cases. The boundary layer is usually not well mixed (there are convective plumes and downdraft regions side by side) so that overlap determination remains a problem. On the other hand, the Polly beam was fixed to an off-zenith angle of 5 degrees. We think, it is always worthwhile to use the opportunity during lidar comparison studies to check the OVP functions of all involved lidars during their normal operation (that means when they look vertically). The proposed method is more easily applicable in our case, and it has been previously applied with accurate results (e.g., Guerrero-Rascado et al., 2010; Sicard et al., 2020).

**R2C2.** The manuscript needs a deep English editing, because some parts are not clear at all. I was editing some parts, but it is not a reviewer role.

Authors' response: We are sincerely grateful for the English editing of the manuscript as performed by the reviewer. An English revision in deep of the manuscript have been also done, and changes are shown in the revised version of the manuscript. But we think (as we know from former publications) that an AMT language editor will check the manuscript as well. Finally, several experienced co-authors will be forced to check the proof-readings and remove 'bad English' phrases.

**R2C3.** Some sections in the manuscript seem to be out of context. As stated in the title and mostly in the abstract, the main objective is to calibrate the MPL instruments with respect to the reference EARLINET lidars. The part where POLIPHON algorithm is applied is not adding value to the paper with respect to its main goal. I suggest to the authors to better contextualize it (maybe editing English will make it clearer) or delete it.

Authors' response: We decided to follow the reviewer #2' suggestion. Hence, corresponding sections (2.4.2 and 3.4), and related comments in the overall text, have been removed.

**R2C4.** Moreover, I think that the retrieval doesn't make so much sense. First, Leipzig is not the best spot to detect dust outbreaks, because the aerosol layer traveled so much before reaching the observation site. Then dividing the backscattering coefficient into those 3 categories is rather audacious and potentially wrong. There is not any information regarding the aerosol size distribution. Then Dc and Df how are assessed ? Just using the Particle Depol Ratio and the LR? In this case, no information is available on how the dust particles aged, i.e. if dust mixes up with urban or continental aerosol. Also, the used values are probably found for some specific measurement campaigns and cannot be assumed valid in general. For this reason, those values will show a very high variability making the error on retrieval huge. What if, during the advection, the dust particles mix with other aerosol particles? The LR changes, the depolarization changes...

Authors' response: We agree that Leipzig is not the best station to observe dust intrusions. However, dust air masses arrived to Leipzig on 29 and 30 June 2019. The dust case study selected in this work was examined in deep in Córdoba-Jabonero et al. (2021), where the arrival of dust particles to Leipzig on 29 and 30 June 2019 was confirmed and analysed by using aerosol travel and forecast modelling, AERONET data and lidar observations. POLIPHON algorithm allowed separating the optical properties of each Dc and Df components in aerosol mixed cases (in particular, see deleted Fig. 8, corresponding to a dust mixed scenario as observed on 30 June afternoon, and also Fig. 3 in Córdoba-Jabonero et al., 2021). POLIPHON is a depolarization-based method (i.e., Mamouri and Ansmann, 2017), which is well validated in a variety of field activities (e.g.,

Genz et al., 2011; Düsing et al., 2018; Mamali al., 2018; Haarig et al., 2019; Marinou et al., 2019) and applied in numerous studies (e.g., Córdoba-Jabonero et al., 2018; Ansmann et al., 2019; Baars et al., 2019; Marinou et al., 2019; Costa-Surós et al., 2020; Georgoulias et al., 2020; Hofer et al., 2020).

**References**

Ansmann, A., Mamouri, R.-E., Hofer, J., Baars, H., Althausen, D., and Abdullaev, S. F.: Dust mass, cloud condensation nuclei, and ice nucleating particle profiling with polarization lidar: updated POLIPHON conversion factors from global AERONET analysis, Atmos. Meas. Tech., 12, 4849–4865, https://doi.org/10.5194/amt-12-4849-2019, 2019.

Baars, H., Ansmann, A., Ohneiser, K., Haarig, M., Engelmann, R., Althausen, D., Hanssen, I., Gausa, M., Pietruczuk, A., Szkop, A., Stachlewska, I. S., Wang, D., Reichardt, J., Skupin, A., Mattis, I., Trickl, T., Vogelmann, H., Navas-Guzmán, F., Haefele, A., Acheson, K., Ruth, A. A., Tatarov, B., Müller, D., Hu, Q., Podvin, T., Goloub, P., Veselovskii, I., Pietras, C., Haeffelin, M., Fréville, P., Sicard, M., Comerón, A., Fernández García, A. J., Molero Menéndez, F., Córdoba-Jabonero, C., Guerrero-Rascado, J. L., Alados-Arboledas, L., Bortoli, D., Costa, M. J., Dionisi, D., Liberti, G. L., Wang, X., Sannino, A., Papagiannopoulos, N., Boselli, A., Mona, L., D'Amico, G., Romano, S., Perrone, M. R., Belegante, L., Nicolae, D., Grigorov, I., Gialitaki, A., Amiridis, V., Soupiona, O., Papayannis, A.,Mamouri, R.-E., Nisantzi, A., Heese, B., Hofer, J., Schechner, Y. Y., Wandinger, U., and Pappalardo, G.: The unprecedented 2017–2018 stratospheric smoke event: decay phase and aerosol properties observed with the EARLINET, Atmos. Chem. Phys., 19, 15183–15198, https://doi.org/10.5194/acp-19-15183-2019, 2019.

Córdoba-Jabonero, C., Sicard, M., Ansmann, A., del Águila, A., and Baars, H.: Separation of the optical and mass features of particle 30 components in different aerosol mixtures by using POLIPHON retrievals in synergy with continuous polarized Micro-Pulse Lidar (PMPL) measurements, Atmos. Meas. Tech., 11, 4775-4795, https://doi.org/10.5194/amt-11-4775-2018, 2018.

Córdoba-Jabonero, C., Sicard, M., López-Cayuela, M.-A., Ansmann, A., Comerón, A., Zorzano, M.-P., Rodríguez-Gómez, A., and Muñoz-Porcar, C.: Aerosol radiative impact during the summer 2019 heatwave produced partly by an inter-continental Saharan dust outbreak. Part 1. Shortwave dust direct radiative effect, Atmos. Chem. Phys., https://doi.org/10.5194/acp-2020-1013, accepted, 2021.

Costa-Surós, M., Sourdeval, O., Acquistapace, C., Baars, H., Carbajal Henken, C., Genz, C., Hesemann, J., Jimenez, C., König, M., Kretzschmar, J., Madenach, N., Meyer, C. I., Schrödner, R., Seifert, P., Senf, F., Brueck, M., Cioni, G., Engels, J. F., Fieg, K., Gorges, K., Heinze, R., Siligam, P. K., Burkhardt, U., Crewell, S., Hoose, C., Seifert, A., Tegen, I., and Quaas, J.: Detection and attribution of aerosol-cloud interactions in large-domain large-eddy simulations with the ICOsahedral Non-hydrostatic model, Atmos. Chem. Phys., 20, 5657–5678, https://doi.org/10.5194/acp-20-5657-2020, 2020.

Düsing, S., Wehner, B., Seifert, P., Ansmann, A., Baars, H., Ditas, F., Henning, S., Ma, N., Poulain, L., Siebert, H., Wiedensohler, A., and Macke, A.: Helicopter-borne observations of the continental background aerosol in combination with remote sensing and ground-based measurements, Atmos. Chem. Phys., 18, 1263-1290, https://doi.org/10.5194/acp-18-1263-2018, 2018.

Genz, C., Schrödner, R., Heinold, B., Henning, S., Baars, H., Spindler, G., and Tegen, I.: Estimation of cloud condensation nuclei number concentrations and comparison to in situ and lidar observations during the HOPE experiments, Atmos. Chem. Phys., 20, 8787–8806, https://doi.org/10.5194/acp-20-8787-2020, 2020.

Georgoulias, A. K., Marinou, E., Tsekeri, A., Proestakis, E., Akritidis, D., Alexandri, G., Zanis, P., Balis, D., Marenco, F., Tesche, M., and Amiridis, V.: A first case study of CCN concentrations from spaceborne lidar observations, Remote Sens.,12, 1557, https://doi.org/10.3390/rs12101557, 2020.

Haarig, M., Walser, A., Ansmann, A., Dollner, M., Althausen, D., Sauer, D., Farrell, D., and Weinzierl, B.: CCN concentration and INP relevant aerosol profiles in the Saharan Air Layer over Barbados from polarization lidar and airborne in situ measurements, Atmos. Chem. Phys. Discuss., https://doi.org/10.5194/acp-2019-466, in review, 2019.

Hofer, J., Ansmann, A., Althausen, D., Engelmann, R., Baars, H., Abdullaev, S. F., and Makhmudov, A. N.: Long-term profiling of aerosol light extinction, particle mass, cloud condensation nuclei, and ice-nucleating particle concentration over Dushanbe, Tajikistan, in Central Asia, Atmos. Chem. Phys., 20, 4695–4711, https://doi.org/10.5194/acp-20-4695-2020, 2020.

Mamali, D., Marinou, E., Sciare, J., Pikridas, M., Kokkalis, P., Kottas, M., Binietoglou, I., Tsekeri, A., Keleshis, C., Engelmann, R., Baars, H., Ansmann, A., Amiridis, V., Russchenberg, H., and Biskos, G.: Vertical profiles of aerosol mass concentration derived by unmanned 35 airborne in situ and remote sensing instruments during dust events, Atmos. Meas. Tech., 11, 2897-2910, https://doi.org/10.5194/amt-11-2897-2018, 2018.

Mamouri, R.-E. and Ansmann, A.: Potential of polarization/Raman lidar to separate fine dust, coarse dust, maritime, and anthropogenic aerosol profiles, Atmos. Meas. Tech., 10 (9), 3403-3427, https://doi.org/10.5194/amt-10-3403-2017, 2017.

Marinou, E., Tesche, M., Nenes, A., Ansmann, A., Schrod, J., Mamali, D., Tsekeri, A., Pikridas, M., Baars, H., Engelmann, R., Voudouri, K.-A., Solomos, S., Sciare, J., Groß, S., Ewald, F., and Amiridis, V.: Retrieval of ice-nucleating particle concentrations from lidar observations and comparison with UAV in situ measurements, Atmos. Chem. Phys., 19, 11315–11342, https://doi.org/10.5194/acp-19-11315-2019, 10 2019.

**R2C5.** Being the P-MPL a product commercially available, it is not possible to establish with precision which technology is used to detect the depolarized laser light, because, as stated on MPLNET website, there exist at least two different P-MPL models that

depend on fabrication year. For the P-MPL models produced before 2013, the use of nematic liquid crystal polarizer introduces a delay in data rates. A new P-MPL model was developed around 2013 following Flynn et al 2007, but using a ferroelectric liquid crystal (FLC) for faster data rates and a slightly modified measurement strategy to accommodate the difference in polarizer properties. For this reason, as long as a proper instrument characterization and stability study of the polarized design and its calibration procedures will be not available, equation 4 and section 3.2 are based on speculations.

Authors' response: Sect. 2.2.1 in the Methodology Section has been modified in order to clarify up to some extent these issues. In particular, the P-MPL version of the 44245 unit uses a ferroelectric liquid crystal (FLC) with a switching time of $\sim 100\ \mu$s. Besides, it was tested by the manufacturer, and a testing report was provided with the P-MPL system. Eq. 4 (now Eq. 2 or 6 in the revised version) is adapted from Flynn et al. (2007) and currently applied for providing the MPLNET version 3 data products (Welton et al., 2018), besides having been applicable in some particular studies (e.g., Sicard et al., 2016; Córdoba-Jabonero et al., 2018; Lewis et al., 2020; Lolli et al., 2020). Sect. 3.2 is based on the experimental analysis performed, indeed, in this work (see also references included).

**R2C6.** Specific comments are found in the attached files.

Authors' response is introduced in the attached supplement pdf file using the Adobe Acrobat tools. Corresponding changes have been also included in the revised version of the manuscript.

---

## Author Response (AR2)

**Authors' response (in blue) to the Reviewer #3's comments:**

The authors thank Reviewer #3 for their comments and suggestions. A general revision of the overall manuscript has been performed, including also Abstract, main text, Conclusions and References. Required changes and modifications have been introduced in the text of the current revised version of the manuscript by using the Word Track Changes tools.

Next, the authors respond to the particular comments of the reviewer #3.

**- Reviewer 3**

**R3C1.** This paper describes a comparison between data collected using the authors Micro Pulse Lidar (MPL) and another lidar from PollyNet. Although the MPL used in this study normally operates as part of the MPLNET, it was removed from its normal station and the data collected during this campaign were processed by the authors with their own processing, calibrations, and data collection techniques. The data collected here are not from MPLNET, nor did they utilize that projects established methodology (including existing calibration process for both polarization and overlap). Therefore the results of this study are not widely applicable, and definitely do not accurately represent the data produced by the MPLNET project. This limits the overall usefulness of this study.

Authors' response: We agree. The polarized Micro Pulse Lidar (P-MPL) data collected in this work are not part of MPLNET collection, as the experimental intercomparison campaign was performed in other different station from the usual MPLNET site of El Arenosillo (Huelva, station Spain) (https://mplnet.gsfc.nasa.gov/data.cgi?site=El Arenosillo), managed by the Instituto Nacional de Tecnica Aeroespacial (INTA), where it usually operates within the network. This was already stated in the manuscript. Particular regular calibrations and signal processing were applied, which are the same as those described by Campbell et al. (2002) and Welton et al. (2002), being within the principal MPLNET publications, and also by Flynn et al. (2007), whose data processing techniques remain also applicable for polarized MPL systems, as indicated in Welton et al. (2018). To our knowledge, those existing MPLNET overlap and polarization calibrations are just available for a few sites, and not applied yet for El Arenosillo site. That is why one of the aims of this work was, indeed, to achieve a particular experimental overlap correction and a polarization assessment of this particular P-MPL system with the information delivered by its manufacturer (Sigma Space Corp.), that is no related to defined MPLNET calibrations. However, authors think the procedure described in this study can be useful to be applied to similar P-MPL systems that cannot account for those stablished MPLNET calibrations yet.

In order to clarify it, Section 2 has been modified (changes are highlighted using the Word Track Changes tools; see, in particular, page 5, lines 165-175 of the updated revised version of the manuscript).

**R3C2.** The authors also present a main conclusion of this paper as an accurate overlap calibration is needed to retrieve aerosol properties from an MPL. This is by no means a new conclusion, this has been known since the MPL was first used in the 1990s. It is also why the MPLNET project has long ago established methods to deal with this issue. The methods used by MPLNET were ignored in this study and not well referenced. The authors do cite Berkoff 2003 which describes the basis of the current MPLNET method to retrieve overlap, however they do not discuss the relevance to their process shown in the paper (which is very similar, involving retrieval of the overlap from the two lidar profiles).

Authors' response: Indeed, every lidar system has to be calibrated by overlap (MPL included). The MPLNET stablished methods for overlap calibration, as those described in Berkoff et al. (2003), were not ignored, but actually, they could not be applied in our particular case. The two procedures for overlap function determination shown in Berkoff et al. (2003) are different from that presented in our work. We presented an alternative experimental method based on the cross-comparison of the backscattered signal recorded by the uncorrected lidar system (our MPL) with respect to that collected by a reference (overlap-corrected) lidar (in this work, EARLINET lidars). A similar methodology has been also used for the overlap correction of other lidars and ceilometers (i.e., Guerrero-Rascado et al., 2010; Sicard et al., 2020; and references therein). We decided to apply this experimental procedure because we wanted to validate the overlap function delivered by the manufacturer and did not have yet the chance to apply the other two overlap calibration procedures described in Berkoff et al. (2003), which have their limitations as well: 1) performing measurements under atmospheric stable and homogeneous conditions with the MPL pointing in horizontal, or 2) making use of a secondary wide field-of-view receiver (WFR) telescope.

Therefore, the main goal was achieved, i.e., determining a well-characterized overlap function for this specific P-MPL system, allowing the retrieval of reliable profiles of aerosol properties.

For clarification, the Introduction has been improved (in particular, page 3, lines 90-98 in the updated revised version of the manuscript.

**R3C3.** In addition, there are a number of errors present in this work and the authors do not present a clear understanding of the polarized MPL. Instead, a very vague and generalized description of the polarization problems is given to describe the causes of bias, for instance the authors say "This offset represents a correction to account for any slight mismatch in the transmitter and detector polarization planes and any impurity of

the laser polarization state", and reference older work from non-MPL polarized lidars (that utilize an entirely different optical design) and an older polarized MPL using a different crystal polarizer. This statement is simply vague and inaccurate. The authors just apply an offset to make the polarized MPL data agree better with the Polly lidar, but appear to have no idea what is causing the bias (or how it might manifest in other ways). If the authors attempt to describe their work as an "experimental assessment of a Micro Pulse Lidar system" then they should have an expert level understanding of the instrument itself. None is presented in this paper. Again, the MPLNET project has established methods for calibrating polarized lidars that were not well referenced in this paper. The authors cite a Welton 2018 paper, but do not discuss the findings from it that bear directly on the polarization calibration of MPL instruments.

Authors' response: As stated before in **R3C2**, we presented an alternative experimental method based on the cross-comparison of the backscattered signal recorded by the uncorrected lidar system (our MPL) with respect to that collected by a reference (overlap-corrected) lidar (in this work, EARLINET lidars). In particular, this methodology has been also used for the overlap correction of other lidars and ceilometers (i.e., Guerrero-Rascado et al., 2010; Sicard et al., 2020; and references therein). This kind of calibration (overlap, polarization, ...) procedures are usually performed in experimental intercomparison campaigns in order to evaluate different instruments in operation within diverse atmospheric networks (i.e., EARLINET, AERONET, NDACC, ...). The experimental performance of the instrumentation involved in such campaigns is assessed this way. A similar concept has been applied in our case. Such a method could be applied in case the MPLNET stablished calibrations are not applicable for a certain instrument due to manifold reasons, but it does not mean that this procedure is aimed to become a standard for MPLNET.

It is out of scope to try to accurately describe the instrument itself in this work. Instead, a relatively brief description of the MPL system used is introduced in the manuscript, since the principal issue is focused on experimental corrections and aerosol properties retrieval. There are already several works presenting a more complete description and operation of the instrument, including MPLNET publications, and specific instrumental manuals as provided by the manufacturer (as indicated in the paper). The P-MPL system of our work corresponds to the MPL version 4B design (v. 4B), which uses a Ferroelectric Liquid Rotator (FLR) instead of a Liquid Crystal Retarder (LCR) (in a previous MPL v.4), with a faster switching time. The signal processing techniques that are described by Flynn et al. (2007) for the LCR version remain also applicable for the FLR one, as stated in Welton et al. (2018). Particular signal processing procedures are applied in our work, which are the same adapted from Flynn et al. (2007). We presented an experimental polarization correction based on real measurements as an alternative, due to the unavailability of the special and specific methods for polarization calibration within MPLNET at the time of the validation campaign. The presented alternative procedure is also used in other works (e.g., Córdoba-Jabonero et al., 2013, and references therein) and represents a fast and reliable method to be used and implemented in our P-MPL system in order to obtain good results on aerosol optical retrievals.

For clarification, Sections 2.2.1, 2.4 and 3.2 have been modified (changes are highlighted using the Word Track Changes tools) in the updated revised version of the manuscript.

**R3C4.** I do not understand why MPLNET is discussed in this paper, nor mentioned in the acknowledgements. Other than the fact that this lidar normally operates in the MPLNET in Spain, this particular study is from a redeployment of the MPL outside of MPLNET using the authors own processing and calibrations. It should be made very clear in this manuscript that the methods described here are not part of MPLNET, nor applicable to MPLNET as they have previously defined methodologies. If the intention is otherwise, then how that would be achieved is not discussed at all. There appear to be no NASA funded contributions to this work, so why is that included in the acknowledgements section? If this did occur it is not obvious in the paper, if not then this should be removed from the acknowledgements.

Authors' response: MPLNET is not discussed in this work. It is just mentioned that the P-MPL of this work is the usual operative system in the MPLNET site of El Arenosillo station (Huelva, Spain). As stated before, P-MPL data used in our work do not appear in the MPLNET database as they were obtained during a particular intercomparison campaign performed in other different place (Leipzig, Germany) from the operative MPLNET El Arenosillo site. The calibration/correction procedures described in our work are not MPLNET standards, but they are used by other lidar/ceilometer community to retrieve accurate results (e. g., Guerrero-Rascado et al., 2008; Sicard et al., 2020). In addition, as the MPL system of our work is the current lidar operating in the MPLNET El Arenosillo site, the following statement was already included in the Acknowledgements section of our work: "The MPLNET project is funded by the NASA Radiation Sciences Program and Earth Observing System.", as the MPLNET policy indicates in the website.

In general, changes are highlighted using the Word Track Changes tools in the updated revised version of the manuscript.

Authors would like again to highlight that the experimental assessment procedure shown in our work is not a MPLNET calibration procedure, because none of the reported MPLNET calibrations (Berkoff et al., 2003; Welton et al., 2018) have been applied yet to the El Arenosillo P-MPL instrument. In addition, data are not MPLNET data, because they were obtained at a different site from the MPLNET El Arenosillo site.

We are really aware of the hard work to implement the calibration procedures in an atmospheric observational network, likewise in MPLNET. Any lidar system needs to be calibrated, and mainly, by overlap and depolarization in order to correct the lidar measurements and obtain good-quality data to be used in research. That was one of the reasons to carry out the intercomparison campaign as described in this work for the experimental assessment of the El Arenosillo MPL system, as stated in the manuscript. As it has not been possible to use the MPLNET calibration procedures by now (we wish

that will be possible in a near future, of course), we used other alternative methods, as those shown in this work, which are based on the intercomparison with reference goodcalibrated lidars (EARLINET lidars in this work) to independently check the performance of that P-MPL system. This is a usual methodology carried out in other intercomparison campaigns of different atmospheric networks (EARLINET, AERONET, NDACC, ...), where the calibration of the instruments for performance testing is carried out in other different places from the regular operative site, as it is in our case. In addition, this is a common benefit, reinforcing the network. In fact, the overlap function as obtained in this work is the current one approved and used for correcting the El Arenosillo MPL data within MPLNET. In this point, we also want to highlight the help from MPLNET in keeping the MPL systems and the data analysis up-to-date, alerting about new problems detected in the measurements and/or monthly calibrations, data missing or any other doubts on the instrumentation. In this sense, we think that acknowledgements to MPLNET are justified. More of those efforts should be done on the way to combine all existing networks in Europe (EARLINET), Asia (AD-NET), Latina America (LALINET) and also MPLNET within the future vision of GAW (Global Atmospheric Watch) Aerosol Lldar **Observations Network (GALION).**

Authors seem to be convinced that the paper can really represent a meaningful contribution, despite it may not be relevant from the point of view of MPLNET. Our goal was to focus on general or basic problems when using lidar measurements for atmospheric profiling in the lower and middle troposphere. And we clearly think the lessons we obtained are very useful. We show and demonstrate how we can obtain the overlap profiles with upward looking lidars (in their exact measuring and monitoring configuration). Our paper is clearly a new and valuable contribution to the lidar literature.

**References**

Berkoff, T. A., Welton, E. J., Campbell, J.R., Scott, V.S., and Spinhirne, J. D.: Investigation of Overlap Correction Techniques for the Micro-Pulse Lidar NETwork (MPLNET), 2003 IEEE International Geoscience and Remote Sensing Symposium (2003 IGARSS). Proceedings (IEEE Cat. No. 03CH37477), 4395-4397, https://doi.org/10.1109/IGARSS.2003.1295527, 2003.

Flynn, C. J., Mendoza, A., Zheng, Y. and Mathur, S.: Novel polarization-sensitive micropulse lidar measurement technique, Opt. Express, 15 (6), 2785-2790, https://doi.org/10.1364/OE.15.002785, 2007.

Córdoba-Jabonero, C., Guerrero-Rascado, J. L., Toledo, D., Parrondo, M., Yela, M., Gil, M. and Ochoa, H. A.: Depolarization ratio of polar stratospheric clouds in coastal Antarctica: comparison analysis between ground-based Micro Pulse Lidar and space-borne CALIOP observations, Atmos., Meas. Tech., 6 (3), 703-717, https://doi.org/10.5194/amt-6-703-2013, 2013.

Guerrero-Rascado, J. L., Costa, M. J, Bortoli, D., Silva, A. M., Lyamani, H., and Alados-Arboledas L: Infrared lidar overlap function: an experimental determination, Opt. Express, 18, 20350-20369, 2010.

Sicard, M., Rodríguez-Gómez, A., Comerón, A., Muñoz-Porcar, C.: Calculation of the Overlap Function and Associated Error of an Elastic Lidar or a Ceilometer: Cross-Comparison with a Cooperative Overlap-Corrected System, Sensors, 20 (21), 6312, https://doi.org/10.3390/s20216312, 2020.

Welton, E. J., Stewart, S. A., Lewis, J. R., Belcher, L. R., Campbell, J. R., and Lolli, S.: Status of the Micro Pulse Lidar Network (MPLNET): Overview of the network and future plans, new version 3 data products, and the polarized MPL, EPJ Web Conf., 176, 09003, https://doi.org/10.1051/epjconf/201817609003, 2018.